# Tor Anonymous Traffic Identification Based on Parallelizing Dilated Convolutional Network

Yunan Lu [1], Manchun Cai [1,*], Ce Zhao [1] and Weiyi Zhao [2]

[1]   College of Information and Cyber Security, People's Public Security University of China, Beijing 100038, China
[2]   Faculty of Engineering, The University of Hong Kong, Hong Kong, China
[*]   Correspondence: caimanchun@ppsuc.edu.cn

**Abstract:** The widespread use of the onion browser (Tor) has provided a breeding ground for the proliferation of cybercriminal activities and the Tor anonymous traffic identification method has been used to fingerprint anonymous web traffic and identify the websites visited by illegals. Despite the considerable progress in existing methods, problems still exist, such as high training resources required for the identification model, bias in fingerprint features due to the fast iteration of anonymous traffic and singularity in the definition of traffic direction features. On this basis, a Tor anonymous traffic identification model based on parallelizing dilated convolutions multi-feature analysis has been proposed in this paper in order to address these problems and perform better in website fingerprinting. A single-sample augmentation of the traffic data and a model combining multi-layer RBMs and parallelizing dilated convolutions are performed, and binary classification and multi-classification of websites are conducted for different scenarios. Our experiment shows that the proposed Tor anonymous traffic recognition method achieves 94.37% accuracy and gains a significant drop in training time in both closed-world and open-world scenarios. At the same time, the enhanced traffic data enhance the robustness and generalization of our model. With our techniques, our training efficiency has been improved and we are able to achieve the advantage of bi-directional deployability on the communication link.

**Keywords:** Tor; anonymous traffic identification; parallelizing dilated convolutions; bi-directional deployability

## 1. Introduction

In response to the large-scale network surveillance and cookie information abuse on the Internet, anonymous communication systems avoid monitoring and network eavesdropping on users' personal information by providing technical methods to hide the content and metadata of communications, which include three main types: anonymous access, anonymous routing and dark web services. The onion browser (Tor), famous for being the most popular of these anonymous communication systems, has garnered thousands of relay nodes and millions of daily users. Tor sets connections through three-hop nodes, transmits data as cells over the link, and uses onion routing to encrypt traffic at all layers. However, Tor is also used by criminals to carry out all kinds of illegal transactions on the dark web, concealing their criminal behaviors, which makes it difficult for public security authorities to investigate and collect evidence. Therefore, the effective identification of anonymous Tor traffic and its corresponding category is crucial to combat dark web crimes.

In website fingerprinting research, fingerprint features are the direct data objects dealt with by identification and defense methods [1]. For the website fingerprinting attack technique, the attacker explores the characteristics of the training data to train a classifier and applies it to the test data to infer the specific category of the website it belongs to. The time, direction and size of the packets transmitted between the client and the remote server are analyzed in order to identify different websites. Currently, Tor website fingerprinting

techniques are applied in both open-world and closed-world attack scenarios. Closed world means that anonymous Tor users only visit a specific number of sensitive web pages, and a web fingerprinting attacker needs to identify the specific sensitive web pages from already known traffic characteristics. An open-world scenario, which is closer to a real web scenario, means that Tor Anonymous users can access a variety of pages through their browsers, and the attacker needs to determine whether the pages they visit are sensitive or not.

In this paper, we propose a Tor anonymous traffic identification model based on a parallelizing dilated convolutional network for the continuity, bi-directionality and frequent iteration of anonymous communication links in Tor anonymous traffic. The main contributions of this paper are as follows:

(1)    In the data processing stage, the directional features are enhanced through cropping and adding data. We then combine the methods used in previous studies to identify anonymous traffic using a single directional feature or timestamp feature.

(2)    In the model training stage, we use parallelizing dilated convolutional networks and multi-layer RBMs to extract features and classify web pages. A multi-core convolutional network parallelizing architecture is built to extract the sequential features of cell packets and the expanded convolutions make it possible to expand the sensory field of the model without pooling operations, enhancing the sensory range of a single cell packet for continuous positive/negative traffic data with neighboring timestamps, compared to the traditional CNN architecture.

(3)    In the model implementation stage, we provide the possibility of bi-directional deployment of the traffic identification model over the Tor communication link and improve the robustness of the model. Experiments are conducted on the collected dataset to compare with the DF model, LSTM model, SDAE model, etc. The experiments show that our method alleviates the problem of insufficient generalization of the model due to irregular updates of web content in real-life situations. The reduced number of parameters speeds up the convergence of the model and the proposed method improves the recognition accuracy compared to previous Tor anonymous traffic identification methods.

The arrangement of the paper is as follows: Section 2 introduces the research related to Tor website fingerprinting attack from similarity discrimination, machine learning and deep learning perspectives, then leads to our model based on the existing research. Section 3 introduces the design of our model. Section 4 presents the environment and dataset details regarding our experiment and verifies the effectiveness of the proposed model in Tor anonymous traffic identification through the experimental results. Section 5 summarizes the work of this paper and proposes directions for future research.

## 2. Related Work

### 2.1. Tor Website Fingerprinting Attack Architecture

The user communicates with the target web page in two stages: the user installs the Tor browser client, accesses the directory server to obtain a list of available relay nodes, selects the three best relay nodes, establishes a link to the target server via these three Tor relay nodes and finally sends and receives packets over the communication link established between the user and the target web page. In theory, two approaches can be used to capture Tor traffic data during the above process. In the first approach, the attacker acts as a Tor ingress node, as only all Tor nodes are transmitting encrypted data and the node closest to the sender is able to read the original unencrypted data. In the second approach, the attacker acts as, for example, an ISP (Internet Service Provider) or an AS (Autonomous System) to capture the traffic between the sender and the Tor entry node. In practical terms, the first approach is infeasible because the Tor protocol randomly selects the Tor nodes to be assigned and it is difficult for a particular sender to connect directly to the attacker's node. The second approach is feasible, which fits our real-world scenario, and Figure 1 represents this way of the traffic capture approach.

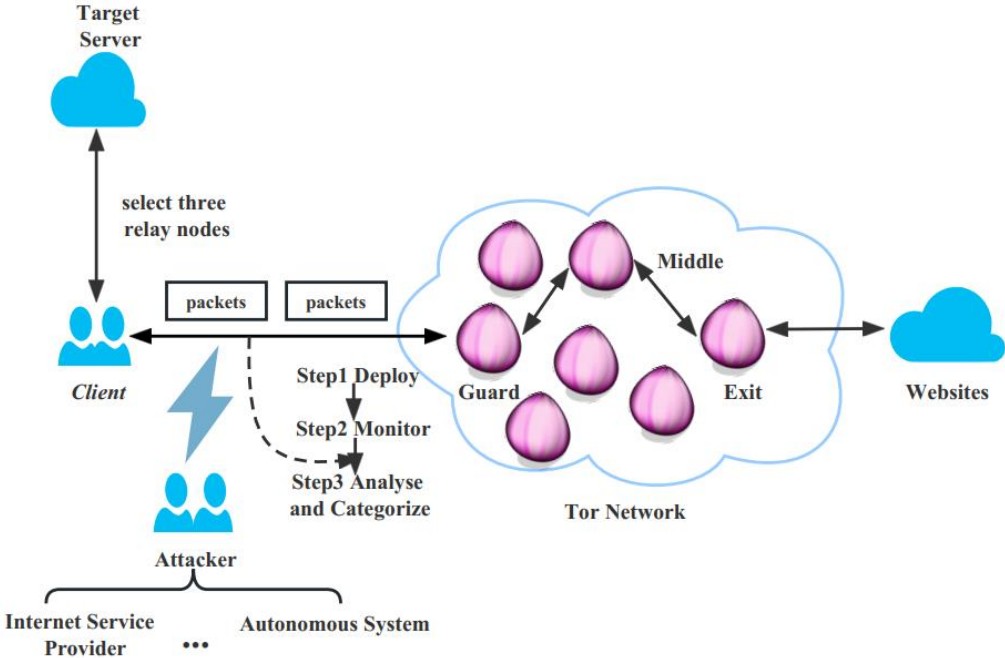

**Figure 1.** Tor website fingerprinting attack architecture.

In this attack architecture, we assume that the attacker is passive, i.e., he can only access the communication link between the user and the Tor network entry node, intercepting the packet information between the client and the Tor network but cannot forward, modify or discard it. At the same time, there is no background traffic in the web page stream, excluding the effect of noisy traffic. The attacker can obtain the IP address of the client through the TLS traffic information on the client and server side, but not the URL of the sensitive web page directly. In this way, the attacker aims to identify the Tor anonymous traffic through the intercepted packet information. Furthermore, to simplify the scenario, we assume that the Tor user only views one web page at a time and does not make any changes to the content.

### 2.2. Overview of Website Fingerprinting Attacks

There are three main types of website fingerprint recognition methods, namely similarity discrimination methods, traditional machine learning methods and deep learning methods [1]. The concept of website fingerprinting attack was first introduced in 2002 by Hintz [2] to address the Safe Web security vulnerability problem, demonstrating the feasibility of obtaining user information through website fingerprinting attacks. Features such as the number of TCP connections, the length of the response message have been used and similarity discrimination methods such as the Jaccard coefficient have been applied [3,4]. The disadvantage is that similarity discrimination methods can cause errors when distinguishing small perturbations of features. The early HTTP protocol leaked features such as resource length, HTML length and total object length, which can be used in identifying static web pages. Therefore, these methods are no longer applicable under the current web technology conditions.

With the further spread of the HTTP 1.1 protocol and the increasing maturity of anonymity techniques, the second-generation onion routing system Tor has gradually become the most widely used anonymous communication system on the web. Machine learning algorithms such as the plain Bayesian algorithm, support vector machine, random forest algorithm and hidden Markov model have been used extensively in website fingerprinting. Panchenko et al. [5] extracted the size, direction and time characteristics of anonymous traffic and ran a support vector machine to classify it, achieving an accuracy of 55% for Tor traffic. Liberatore et al. [6] generated fingerprints using only the length of

messages generated during web access. They used Naïve Bayes as a classifier and finally achieved 73% accuracy. Shahbar et al. [7] used four machine learning methods, namely Naïve Bayes, Bayes Net, C4.5 and random forest, to demonstrate good experimental results in a closed-world environment but poor results were found in an open-world environment at both the circuit level and the flow level. Cai et al. [8] proposed an effective anonymity network traffic identification model called Anon, combining improved mutual information and random forest algorithms to quickly filter irrelevant and redundant features, using the XGboost model for training, achieving better results than previous models. Wang et al. [9] proposed to use Tor cells instead of TCP/IP packets for a better understanding of data and achieved 91% classification accuracy in the closed world and over 95% recall in the open world by comparing experiments to remove the interference of "SENDME" cells in the features and using a traffic instance distance metric in SVM. In sum, the Tor anonymous traffic identification using traditional machine learning methods has generally achieved good results, but the following problems still exist: (1) using the Naive Bayesian classifier as an example, due to the attribute independence requirements of the Naive Bayesian classifier, the dependence between the features selected by each study becomes one of the reasons that affect the performance of the classifier and this limits our feature selection; (2) the complex types of features used lead to more feature extraction workload and there is a problem that the complex types of features used instead lead to low recognition accuracy.

Hinton et al. [10] provided the opening introduction to the concept of deep learning in 2006, and since then deep learning methods have become a popular area for machine learning research, taking advantage of its ability to automatically extract features from raw data. Three main types of deep learning methods used in website fingerprinting include the following: classification methods based on stacked denoised autoencoder, recurrent neural network classification methods based on temporal features and methods based on convolutional neural networks. Convolutional neural network-based classification methods are divided into two categories, including one-dimensional convolutional which is used on one-dimensional traffic data and two-dimensional convolutional which converts network traffic into two-dimensional images for input into a convolutional neural network. Abe et al. [11] used deep learning methods for the first time to perform website fingerprinting attacks, using only cell orientation features and applying stacked denoised autoencoder to reduce the dimensionality of the data vector, reducing the risk of model overfitting and improving accuracy. He experimented with the highest accuracy achieved by both three-layer and two-layer stacked denoising autoencoders in a closed world of 88%, but the attack model is not pre-trained and both cell orientation data and labels are used as input for classification, leaving room for further improvement in the scalability of the model. Rimmer et al. [12] used the LSTM (Long Short-Term Memory) temporal network structure for reference in the fingerprint attack model building and used only the first 150 cells of each traffic data as input data in order to reduce the time complexity of their model, and finally achieved an accuracy of 94.02% in a closed world. The use of the LSTM model solves the RNN long-term dependency problem, which essentially takes into account the existence of a certain temporal relationship between packets communicated during the access to the same web page, and thus also indirectly argues for the feasibility of recurrent neural networks in website fingerprinting attacks. Bhat et al. [13] proposed a VAR-CNN method for website fingerprinting attacks based on ResNet, combining cell timestamp, direction and seven cumulative statistical features to achieve the purpose of using a weak prediction of cumulative statistical features to strengthen the strong prediction of neural networks, and finally reduce the impact of a possible data overshooting problem in anonymous traffic by smaller training samples. Wang et al. [14] proposed a 2ch-TCN approach, which exploited the properties of TCN (temporal convolutional network) to effectively address the gradient explosion/disappearance problem by using dual channels for cell directional features and timestamp features processing, and achieved an accuracy of 93.73% in a closed world. Ma et al. [15] proposed a DBF model for an attack, using burst features alone as unique features and three modules, namely the serial burst

extraction module, burst abstract learning module and burst depth analysis module to effectively improve the model performance. Sirinam et al. [16] proposed a CNN-based method for website fingerprinting (deep fingerprinting) which uses cell orientation as the feature input. A one-dimensional convolutional architecture with eight layers and two fully connected layers was constructed for classification, adding multiple Maxpooling layers during convolution and using 0.1 and 0.5 dropout rates in the convolutional and fully connected layers, respectively. They reduced the loss of data information by ELU and RELU functions at the output level, eventually achieving better results in both closed-world and open-world scenarios. However, the model network structure is too deep and the number of parameters is too large, which results in a long training time and affects its further extension and update. In sum, the accuracy of the Tor anonymous traffic identification method based on deep learning has been greatly improved and there is no need to manually select features, but the following problems still exist: (1) The structure of most models is too deep, which results in the large amount of parameters and too long training time. The high requirements for model training resources lead to slow model training and difficulty in actual use, which affects its further expansion and update. (2) Most of the existing studies have been based on the stability of the network environment in which the traffic data are collected. On the one hand, the Tor anonymous communication dataset is difficult to collect and the available directional data collected needs to be strictly specified in both positive and negative directions. On the other hand, even if it is possible to collect thousands of data from a single web page at a specific period of time to build a website fingerprint, there is inevitably the problem of data obsolescence, i.e., changes in the content of the web page will change the actual website fingerprint characteristics, leading to errors in recognition. Therefore, this paper addresses these issues by augmenting the dataset with cell directional features, providing the possibility of bi-directional deployment of anonymous recognition models over Tor communication links, and effectively extending the self-coding work from Abe [11]. We adopt the burst features that have worked well in the DBF model [15] and use a multi-layer Boltzmann machine to pre-train a total of eight self-extracted features for the self-coding work. To address the problem of a high training resource demand and too long of a training time in deep fingerprinting [16], we use the parallelizing structure and streamlined network structure to reduce the number of parameters required for the model and improve the convergence speed of the model. This improves the sensing range of a single cell packet for a continuous sequence of positive/negative streams with neighboring timestamps, thus effectively balancing model accuracy, robustness and generalization.

## 3. Model Design of This Paper

In this paper, in order to achieve a lightweight framework, a faster model fitting speed and bi-directional deployment of traffic recognition, we perform manual and automatic feature extraction on the intercepted cell units, using multi-layer RBM self-encoding and parallelizing dilated convolutions, respectively, to build the model. In terms of data enhancement, taking into account the characteristics of the Tor link and the bi-directional characteristics of the captured direction data, the crop and add operations are respectively performed. The mental map gives a brief illustration of our design in Figure 2.

### 3.1. Tor Anonymous Traffic Identification Architecture

In this section, we describe the overall architecture of the model and the Tor anonymous traffic identification process, and then we describe the details and advantages of the parallelizing dilated convolutional network that we have employed.

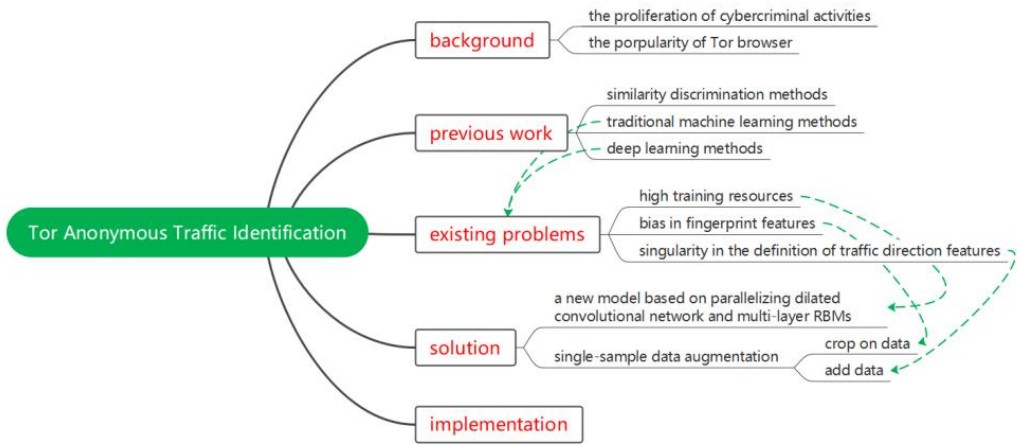

**Figure 2.** Mental map of the design.

3.1.1. Overall Architecture and Tor Anonymous Traffic Identification Process

The core idea of the Tor anonymous traffic recognition model proposed in this paper is to use a combination of manual and automatic feature extraction. Based on single-sample data augmentation in the cell direction, we apply a multilayer restricted Boltzmann machine (RBM) to the manual features for feature parsimony, fusing Tor anonymous traffic multidimensional features and mapping the model output to the corresponding class labels in both binary and multi-classification scenarios through parallelizing one-dimensional dilated convolutional networks and two-dimensional convolutional neural networks, which are used to map the model output to the corresponding class labels. Our architecture includes the Tor traffic data pre-processing module, data encoding module, data enhancement module, multi-feature fusion and training module. The overall architecture is shown in Figure 3. More detail concerning our architecture will be further elaborated on in the text (for each subsequent figure marked in Figure 3).

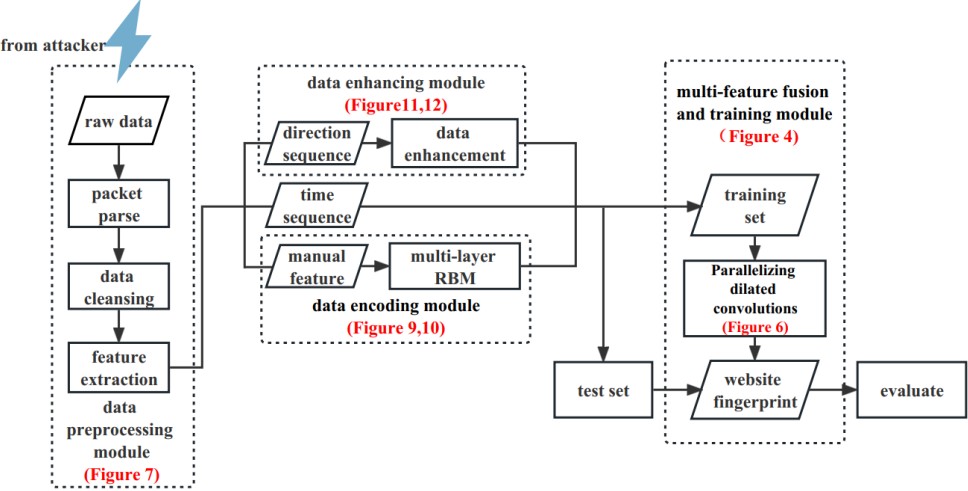

**Figure 3.** Tor anonymous traffic identification architecture.

In the multi-feature fusion and training module, the Tor anonymous traffic recognition multi-classifier and the binary classifier are constructed and combined according to the needs of two different scenarios including closed-world and open-world scenarios. As is shown in Figure 4, we divide the website tags into two-classes and multi-classes (100 classes) and after training we obtain two classifiers, namely binary classifier and multi-classifier. We first discriminate whether the anonymous traffic is sensitive or not through the binary classifier. If not, it is determined as non-sensitive web pages, otherwise the multi-classifier is then used to identify specific sensitive web pages.

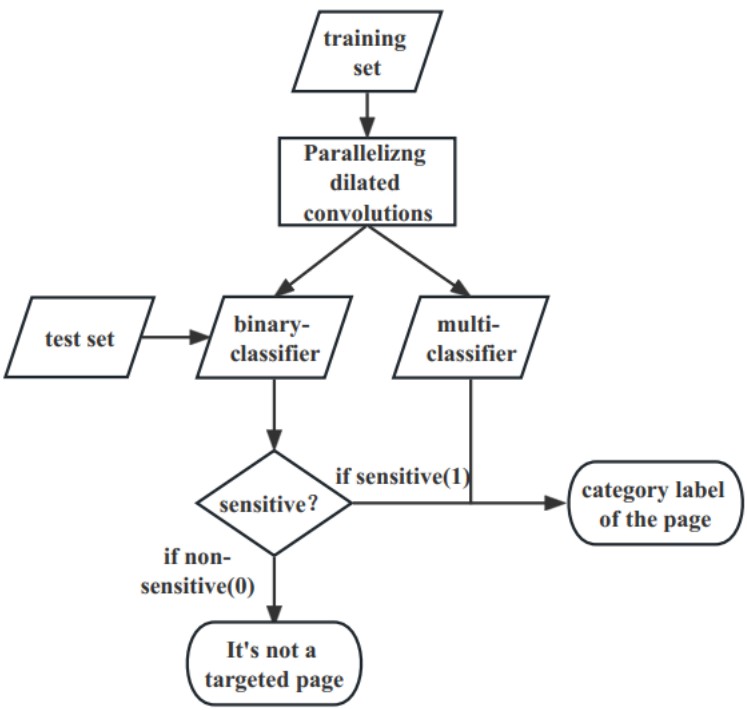

**Figure 4.** Detailed classification of Tor anonymous traffic.

3.1.2. Parallelizing Dilated Convolutions Networks

In the Tor anonymous traffic identification process, timestamp features and directional features are treated as two feature sequences. For each anonymous traffic of client–web server interaction, cell directional features are a sequence of +1, −1 over the communication time range corresponding to each timestamp change, and thus our proposed network architecture requires the extraction of high-level features from both sequences that correspond to our classification task. The LSTM temporal network architecture proposed by Rimmer et al. [12] takes full account of the sequence feature extraction capabilities of their respective models when dealing with sequential problems but still suffers from some gradient disappearance problems and high model training resource requirements as bottlenecks. We construct the convolutional neural network from the perspective of the feature perceptual field, which is the size of the region on the input map corresponding to a point on the feature map at the output of each layer of the convolutional neural network, as shown in Figure 5. The growth in depth increases the number of layers and the growth in width increases the number of neurons in each layer, thus providing a better fit of the model to the training data. However, the resulting increase in the number of network parameters consumes the computational power of the machine to a large extent and increases the risk of overfitting. Previous work has used Maxpooling layers to perform parameter parsimony but there exists a problem of partial loss of sequence information.

We introduce the dilated convolutions proposed by Yu [17] into our model. The dilated convolutions increase the perceptual field of the convolution kernel while keeping the number of parameters constant, so that each convolution output contains a larger range of information, while it ensures that the size of the output feature map remains constant. In two-dimensional convolution, a $5 \times 5$ convolution kernel with a dilation of 2 has the same field of perception as a $5 \times 5$ convolution kernel, but with only nine parameters, a 64% reduction in the number of parameters compared to a $5 \times 5$ convolution. At the same time, we also use the concept of the Inception block proposed by Christian et al. [18] in GoogleNet for reference, combining convolutional layers of different convolutional kernel sizes through parallelism, and stitching the input direction and timestamp data of anonymous traffic through three one-dimensional convolutional layers with dilation coefficients of 1, 2 and 3 to form a deeper matrix in the dimension of the channel, while

performing dimensionality reduction on the original data matrix and subsequently, two layers of two-dimensional convolution are performed. After the flattening operation, the data are mapped to the corresponding classification totals by full concatenation. The convolutional layer parameters use a dropout rate of 0.2 to reduce the complex co-adaptation relationships between neurons, preventing the situation where some features are only effective in other specific feature settings. In this way, the network is forced to learn more robust features, which in turn helps to solve the overfitting problem. In addition, we use the Linear layer, Leaky ReLU activation function and Layer Normalization [19] to add the self-extracted features of the three-layer stacked Boltzmann machine (RBM) to the output of parallelizing dilated convolutions, so as to further use the SoftMax layer to map model outputs to corresponding class labels. The parallelizing dilated convolutions layer is shown in Figure 6, and we also use the Leaky ReLU activation function after each layer of convolution. The Leaky ReLU activation function is a variant of the ReLU activation function, the output of which has a small slope to negative inputs. Since the derivative is always non-zero, this can reduce the appearance of silent neurons, thus solving the problem that the ReLU function enters the negative interval and causes the neurons to be unable to learn. After our experiments, the Leaky ReLU activation function outperforms ReLU, ELU, Sigmoid, SoftMax and other activation functions and best fits the actual scenario.

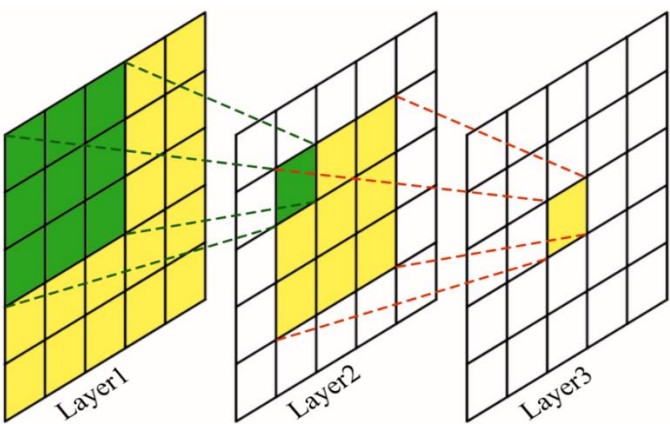

**Figure 5.** Receptive field.

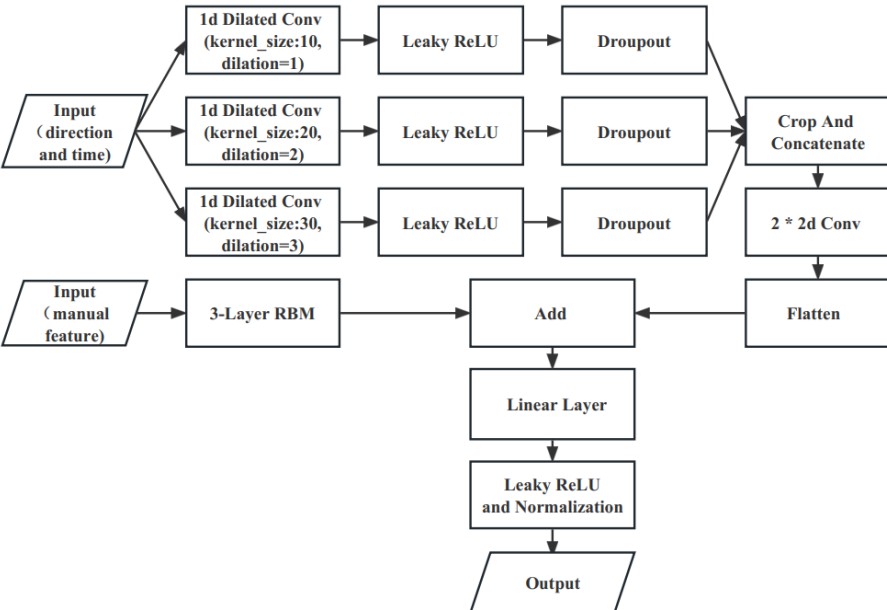

**Figure 6.** Parallelizing dilated convolutions networks.

### 3.2. Tor Traffic Data Pre-Processing

The TCP connection is the basis for the Tor anonymous protocol. On top of the TCP connection runs the TLS transport layer encryption protocol, which ensures that the physical connection is passable and the Tor network is externally encrypted, while Tor divides the encrypted data into cells in units of 512 bytes. Figure 7 depicts this three-layer structure of cell, TLS and TCP.

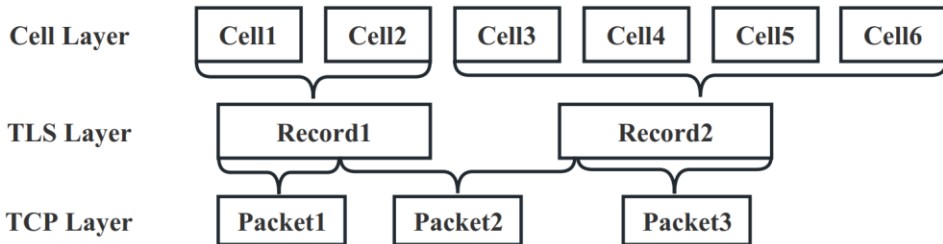

**Figure 7.** Layers of data transport in Tor.

In the data pre-processing process, drawing on the comparison of the effectiveness of the cell, TLS and TCP layer data used in the identification of anonymous traffic [14], we discard the TLS and TCP layer traffic data and use the cell layer traffic data. In this case, each instance includes the direction sequence of the cell and the timestamp sequence obtained by each cell. The intercept part of data in one instance is shown in Figure 8.

| timestamp | drection |
|:---:|:---:|
| 0 | 1 |
| 0 | 1 |
| 0.189541101 | 1 |
| 0.644330025 | -1 |
| 0.644330025 | -1 |
| 0.868875027 | -1 |
| 0.868875027 | -1 |
| 0.903809071 | -1 |
| 0.903809071 | -1 |
| 0.903809071 | -1 |
| 1.020455122 | -1 |
| 1.020455122 | -1 |
| 1.115892172 | -1 |
| 1.125641108 | 1 |

**Figure 8.** Example of dataset.

The first column indicates the sequence of timestamps obtained by the cell, with the first cell sent at time 0. The second column indicates the direction of the cell, with 1 indicating a cell sent from the client to the ingress node Guard and $-1$ indicating a cell sent from the ingress node Guard to the client. Thus, the cell direction sequence, as well as the timestamp sequence can be represented as follows, respectively, in Equations (1) and (2). Then, they are further applied as inputs both in manual feature extraction and automatic feature extraction.

$$X1 = (0, 0, 0.189, 0.644, 0.644, 0.869 \ldots) \tag{1}$$

$$X2 = (1, 1, 1, -1, -1, -1 \ldots) \tag{2}$$

A web page tag is an identifier of a web page category and the classification mark for anonymous traffic recognition. In a closed world, for the 100 different web pages we classify, since the web page labels are independent of each other, we use one-hot coding to encode the web page labels as one-dimensional vectors that can be processed by the neural

network, with each one-dimensional vector corresponding to one in the set of sensitive web pages. For example, one sensitive web page labeled as 0 (web pages are labeled as 0–99, totally 100 web pages in our experiment) is shown in Equation (3), where we apply CrossEntropyLoss between this actual label and our supposed category label to form the Loss Function in our training.

$$W = 1, 0, 0, 0 \ldots, 0 \tag{3}$$

In the open world, we divide all 18,000 instances into two main categories, sensitive web pages for 1 (9000 instances) and non-sensitive web pages for 0 (9000 instances), and the set of web page tags for class II tags. The number of cells in each instance varies because of the varying length of client–server communication in each instance and the different sizes of web traffic instances. Since the model uses batch data, we perform interception and padding operations on the batch data. We set the sequence length to a fixed value of 2000 and if a single instance is less than that sequence length, then we perform a fill operation, and vice versa, we perform an intercept operation to a length equal to the sequence length.

In addition to automatically extracting features from directional and temporal data using parallelizing dilated convolutions networks, drawing on Bhat et al. [20], we manually extract eight cumulative features, combining the burst features used in [11], where a burst is the length of a sequence of directionally identical streams with multiple (more than 2) consecutive occurrences. For example, the length of the +1 stream sequence for the first three timestamps in Figure 8 is a positive burst (length = 3). The eight cumulative features are shown in Table 1.

**Table 1.** Eight Manual Features.

| Number | Manual Feature |
| --- | --- |
| 1 | The number of cells |
| 2 | The number of incoming cells (+1) |
| 3 | The number of outcoming cells ($-1$) |
| 4 | The number of bursts |
| 5 | The number of forward bursts |
| 6 | The number of backward bursts |
| 7 | Total transmission time |
| 8 | Average time to send each burst |

All manual features are encoded in a multi-layer RBM and then added to the results of the parallelizing dilated convolutions networks after the flatten operation, using further operations such as Linear layer, Leaky ReLU and Normalization.

### 3.3. Multilayer Restricted Boltzmann Machine Coding Module

The multi-layer restricted Boltzmann machine is used to encode the input utilizing unsupervised learning. The single-layer RBM is first introduced which consists of two layers of neurons, a layer called the visible layer used for the input training data and a layer called the hidden layer used to extract features; hence, the name of the hidden element is also known as the feature detector. The structure of a single-layer RBM is shown in Figure 9. Since only the inter-layer is symmetrically fully connected within the visible and hidden layers, the neurons within the layers are not interconnected, so given the values of all the visible elements, the values of each hidden element are not correlated with each other. Similarly, given all the hidden elements, the values of the visible elements are not correlated so that parallel computing can be applied to the calculation of the values of each neuron. We process the extracted cell timestamp data (for example in Figure 8, "subtraction" are performed on 0, 0, 0.1895 of the first three timestamps to obtain 0, 0-0, 0.1895-0), and then use it as v1, v2, v3 . . . aa input to the multi-layer RBMs. h1, h2, h3 . . . in the following layer to demonstrate the results obtained after a single-layer encoding.

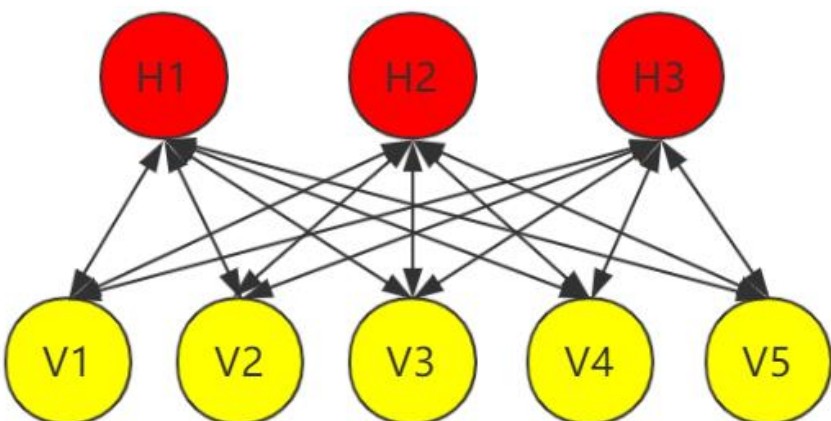

**Figure 9.** Single-layer RBM.

For a Boltzmann machine, the overall energy of the network can be expressed as

$$E(v,h) = -(v^T \cdot W \cdot h + \frac{1}{2} v^T \cdot L \cdot v + \frac{1}{2} h^T \cdot J \cdot h \tag{4}$$

The joint density probability function of $v$ and $h$ [21]:

$$P(v,h) = \frac{e^{-E(v,h)}}{Z} \tag{5}$$

where $Z = \sum_v \sum_h e^{-E(v,h)}$, the explicit layer neuron's bias coefficient matrix $L = [L_{ij}]_{N_v \times N_v}$, the bias coefficient matrix of the hidden layer neuron itself $J = [J_{ij}]_{N_h \times N_h}$, the weight matrix of the neuron connections between the explicit and implicit layers $W = [W_{ij}]_{N_v \times N_h}$ and $N_h$, $N_v$ represents the number of neurons in the hidden layer and the number of neurons in the dominant layer, respectively.

In a constrained Boltzmann machine, the hidden layer neuron $h_j$ is activated by a neuron in the dominant layer $v$. The probability of activation:

$$P(h_j|v) = \sigma\left(b_j + \sum_i W_{ij} v_i\right) \tag{6}$$

where $b_j$ is the bias coefficient of the dominant layer neuron. Since the restricted Boltzmann machine is bi-directionally connected, the explicit layer neuron $v_i$ can equally be activated by the hidden layer neurons activated with a probability of

$$P(v_i|h) = \sigma(c_i + \sum_j W_{ij} h_j,) \tag{7}$$

where: $c_i$ is the bias coefficient of the hidden layer neuron and $\sigma$ is the Sigmoid function, i.e., $\text{Sigmoid}(x) = \frac{1}{1+e^{-x}}$. In the RBM, neurons are independent in the same layer, so the probability density also satisfies independence, the following two equations are obtained:

$$P(h|v) = \prod_{j=1}^{N_h} P(h_j|v) \tag{8}$$

$$P(v|h) = \prod_{i=1}^{N_v} P(v_i|h) \tag{9}$$

When the data are fed into the explicit layer, the RBM calculates the probability of each hidden layer neuron being switched on $P(hj|v)$, taking a random number $\mu$ of 0–1 as the threshold, where neurons greater than this threshold are activated, otherwise they are not. The same is carried out for the explicit layer calculated from the hidden layer. In the learning process of RBM, the contrast scatters algorithm is used to train each piece of traffic data [22]. The input is assigned to the explicit layer, and the probability of each

neuron in the hidden layer being activated ($P(h_j|v)$) is calculated using Equation (5), using Gibbs sampling from the probability distribution to draw a sample $h_1 \sim P(h_j|v)$, and then reconstructing the explicit layer by $h_1$, i.e., calculating the probability of each neuron ($P(v_i|h_1)$) being activated in the explicit layer using Equation (8).

Similarly, a sample is drawn from the calculated probability distribution by using a Gibbs sample $v_2 \sim P((v_i|h_1)$, calculating the probability of each neuron in the hidden layer being activated by $v_2$ to obtain the probability distribution $P((h_2|v_2)$. Then, the weights are updated.

$$W \leftarrow W + \lambda(P(h_1|v_1)v_1 - P(h_2|v_2)v_2) \tag{10}$$

$$b \leftarrow b + \lambda(v_1 - v_2) \tag{11}$$

$$c \leftarrow c + \lambda(h_1 - v_2) \tag{12}$$

After multiple rounds of training, the parameters of the single-layer RBM are updated. The multi-layer RBM refers to the stacking of multiple single-layer RBMs, where the hidden layer of the previous RBM is used as the visible layer of the current RBM as input by iteration to train the hidden layer of the current layer. A three-layer stacked RBM is shown in Figure 10.

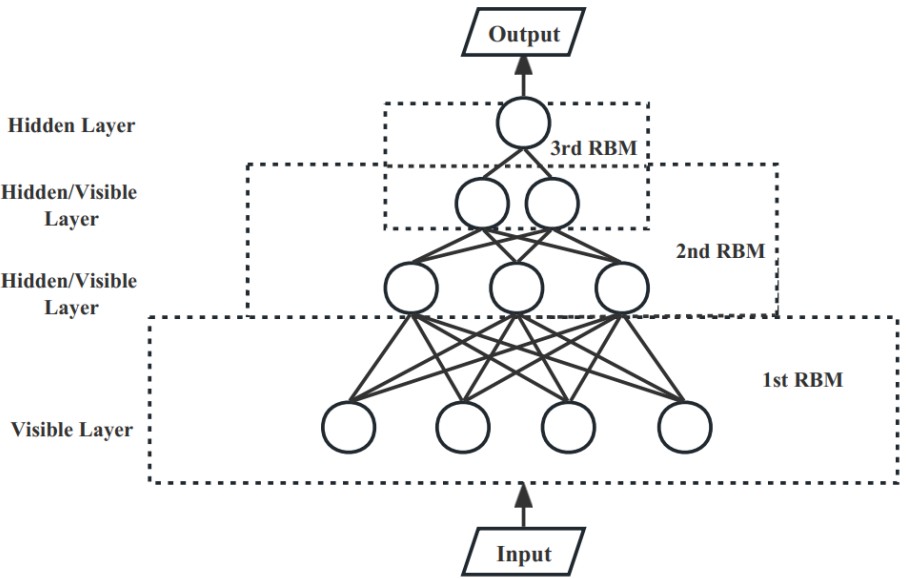

**Figure 10.** Multi-layer RBM.

The multi-layer RBM is trained using an unsupervised greedy layer-by-layer approach, where the training process requires the RBM of the previous layer to be fully trained before the RBM of the current layer can be trained, allowing for better extraction of input data features through a multi-layer architecture. In this paper, the eight manually extracted features are first mapped by the SoftMax function [11] and then fed into a multi-layer RBM for encoding, compensating for the lack of feature extraction capability of single-layer feedforward neural networks.

### 3.4. Data Enhancement and Multi-Feature Fusion Module

On the one hand, as the attacker of the website fingerprinting attack, the link we monitor from the client server to the Tor entry node is updated every 10 min, in which situation there may be mixed traffic. A certain timestamp that we collect may not be the moment when the client server and the Tor entry node end the communication; instead, it may be that the attacker expects the monitoring link to end and monitors the communication of other web traffic on the monitored link. For this reason, we adopt the crop method to remove the monitoring to one-fifth of the data at the end of the traffic to avoid this possible problem, as shown in Figure 11.

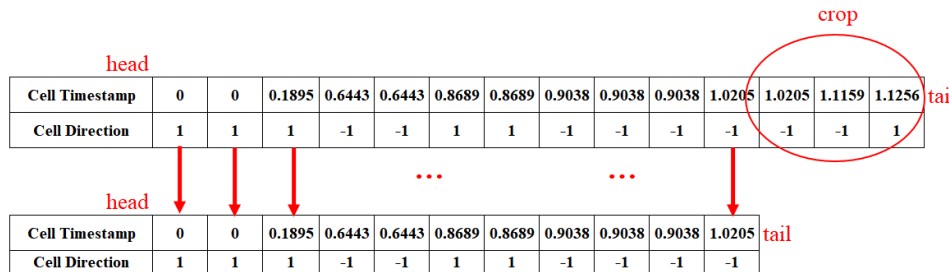

**Figure 11.** Data augmentation (crop on data).

On the other hand, Tor anonymous traffic data are difficult to collect and it is vulnerable to missing data. The size of the data we use to fingerprint a website is often insufficient, so we consider a situation where we use +1 for a cell sent from the client to the ingress node Guard and −1 for a cell sent from the ingress node Guard to the client in the original dataset, where directional features of +1, −1 only represents the direction. We can take the negative operation for all directional features, as shown in Figure 12. From the perspective of the attacker, it is equivalent to changing the location of the user client and the target website to collect the Tor anonymous traffic again. After the negative operation, −1 direction means a cell from the client to the entry node Guard, +1 direction means the opposite. This eliminates the need for the attacker to consider which side of the communication link is strictly positive when performing a website fingerprinting attack, thus providing the possibility of deploying the model in both directions, doubling the amount of data, improving the generalization capability of the Tor anonymous traffic identification model and avoiding overfitting during the training process.

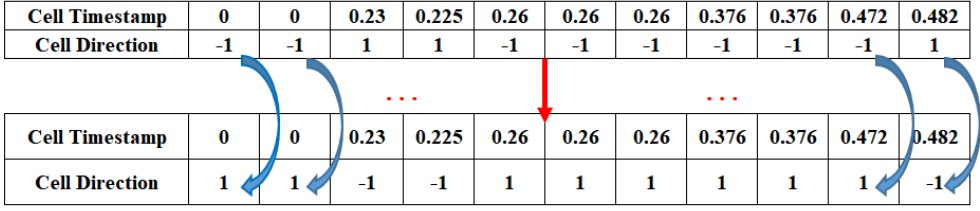

**Figure 12.** Data augmentation (add data).

## 4. Experimental Analysis

The model was implemented using the PyTorch deep learning framework. The experimental environment is shown in Table 2.

**Table 2.** Experimental Environment Parameters.

| Configuration | Parameters |
|---|---|
| OS | Linux64 (Ubuntu 16.04.7 LTS) |
| GPU | NVIDIA GeForce GTX 1080 Ti |
| Video memory | 16 GB |
| Python | 3.8.13 |
| PyTorch | 1.8.1 |
| CPU | Intel(R) Xeon(R) CPU E5-2650 v4 @2.20GHz |
| Memory | 128 GB |

### 4.1. Dataset Partitioning and Hyperparameter Selection

The dataset we have chosen is from Wang et al. [11], collected in 2014 (referred to later as the Wang dataset), and includes 100 sensitive pages with 90 instances each, for a total of 9000 instances, compiled from a list of blocked pages from China, the UK and Saudi Arabia. The dataset includes pages ranging from adult content, seed trackers and social media to sensitive religious and political topics. An additional 9000 non-sensitive pages were

selected from Alexa's top 10,000, excluding pages that intersected with the list of sensitive pages based on the domain name, with one instance per page, for a total of 18,000 instances of sensitive and non-sensitive pages combined. The Wang dataset has been used in many previous studies with good results [11,14,17]. The total number of instances of sensitive and non-sensitive pages is 18,000. As described in the previous section, we divided it into the training set, validation set and test set in the ratio of 8:1:1 with no overlap and random division between the training set, validation set and test set. In the closed world, a total of 9000 sensitive web pages were used, of which 7200 were used for the training set, 400 for the validation set and 400 for the test set. The data augmentation is performed on the training set data, which is expanded from 7200 to 14,400 after directional augmentation, and furthermore, from 14,400 to 28,800 after augmentation by crop operation. The parameters of this model are set as shown in Table 3.

**Table 3.** Hyperparameters Configuration.

| Name of Hyperparameters | Range | Size |
| --- | --- | --- |
| Input dimension | 500, 1000, 1500, 2000, 3000, 5000 | 2000 |
| Optimizer | Adam, Adamax, RMSProp, SGD | Adam |
| Number of RBM layers | [2,5] | 3 |
| Learning rate | 0.0005, 0.001, 0.002, 0.005, 0.01 | 0.002 |
| Training Epoch | [30,200] | 100 |
| decay | [0,0.8] | 0 |
| Hidden units of RBM | [5,30] | 20, 10, 5 |
| Number of out_channels | [5,15] | 10 |
| Dilation coefficient | [1,10] | (1, 2, 3) |
| Kernel_size(Conv1d) | [5,20] | 10, 20, 30 |
| Kernel_size(Conv2d) | [1,30] | (5, 10), (1, 1) |
| Activation function | Sigmoid, ReLu, Leaky ReLu, tanh | Leaky ReLU |
| Dropout | [0.05,0.5] | 0.2 |
| Batch_size | [16,24,32,64,128] | 32 |

*4.2. Assessment Indicators*

In a closed world, model performance is measured using accuracy, loss curves, model training time, etc. In the open world, TPR (true positive rate), FPR (false positive rate), precision, accuracy, F1-Score, etc. are used as metrics:

$$\text{TPR} = \text{Recall} = \frac{\text{TP}}{\text{TP} + \text{FN}} \tag{13}$$

$$\text{FPR} = \frac{\text{FP}}{\text{TN} + \text{FP}} \tag{14}$$

$$\text{Precision} = \frac{\text{TP}}{\text{TP} + \text{FP}} \tag{15}$$

$$\text{Accuracy} = \frac{\text{TP} + \text{TN}}{\text{TP} + \text{TN} + \text{FP} + \text{FN}} \tag{16}$$

$$\text{F1} - \text{Score} = 2 * \frac{\text{Precision} * \text{Recall}}{\text{Precision} + \text{Recall}} \tag{17}$$

where TP is the total number of samples correctly classified as sensitive websites, TN is the total number of samples correctly classified as non-sensitive websites, FP is the total number of samples of websites misclassified as sensitive and FN is the total number of samples of websites misclassified as non-sensitive.

*4.3. Closed-World Performance*

In this section, we train on the Wang dataset using unenhanced data to compare our anonymous traffic recognition model with state-of-the-art approaches in terms of accuracy: the CUMUL proposed by Panchenko et al. [23], SDAE and CNN by Rimmer et al. [12],

deep fingerprinting by Sirinam et al. [16], Var-CNN by Bhat et al. [13] and 2ch-TCN by Wang et al. [14].

As can be seen from Table 4, the Tor anonymous traffic identification method we use achieves an accuracy improvement over state-of-the-art approaches by using parallelizing dilated convolutions for multi-featured large field of view extraction.

**Table 4.** Accuracy On Different Approaches (unaugmented Wang dataset).

| Name of State-of-the-Art Approaches | Accuracy |
|---|---|
| Panchenko-CUMUL | 91.38% |
| Rimmer-CNN | 71.43% |
| Rimmer-SDAE | 87.78% |
| Rimmer-LSTM | 91.11% |
| Deep fingerprinting | 91.38% |
| Bhat-Var-CNN-dir | 93.20% |
| Bhat-Var-CNN | 93.33% |
| Wang-2ch-TCN | 93.73% |
| **Our model** | **94.37%** |

To demonstrate the feasibility of deploying our proposed model in both directions over the communication link, we trained the model with unenhanced data and used the data with the cell direction reversed as the test set. While on the other hand, we trained the model with enhanced data and used the data with the cell direction reversed as the test set. Accuracy and loss for the two scenarios are shown in Table 5.

**Table 5.** Testing Results On Different Trained Models (Wang dataset).

| | Accuracy | Loss |
|---|---|---|
| **Direction-Augmented data model** | **93.17%** | **0.468919** |
| Unaugmented data model | 3.78% | 10.002669 |

The results show that the model trained on cell directionally enhanced data achieves high recognition accuracy for reverse-captured data, while the model trained on unenhanced data has a lower recognition rate for reverse-captured data. This provides the possibility of implementing a bi-directional deployment of the Tor anonymous traffic recognition model.

Next, we combined the cell direction enhancement with the crop approach enhancement and compared the accuracy curves of the model trained on the combined data with the accuracy curves of the model trained on the unenhanced data, as shown in Figure 13.

Where tain_acc represents the classification accuracy of the model on the training dataset during each training round, val_loss represents the classification accuracy of the model on the validation set. The four curves represent the variation curves of train_acc and val_acc with training rounds for data without data enhancement; the variation curves of train_acc and val_acc with training rounds for data enhanced by both methods. We introduce the indicator error rate "abs" to represent the ability of the training model to fit the new data.

$$\text{abs} = |train\_loss - val\_loss| \tag{18}$$

The results show that the two training models achieve essentially equivalent accuracy on the validation set after the models have achieved fit; at the same time, the convergence time of the model trained on the unenhanced data (convergence at 36 EPOCH) is slightly longer than that of the unenhanced data (convergence at 20 EPOCH), but its abs value is approximately twice that of the enhanced data. This suggests that the model trained on the augmented data by both cell direction and crop operations does not reduce the test accuracy while sacrificing some model convergence speed for better fitting of the newly captured anonymous traffic data, avoiding the possibility of overfitting the model and enhancing the robustness of the model.

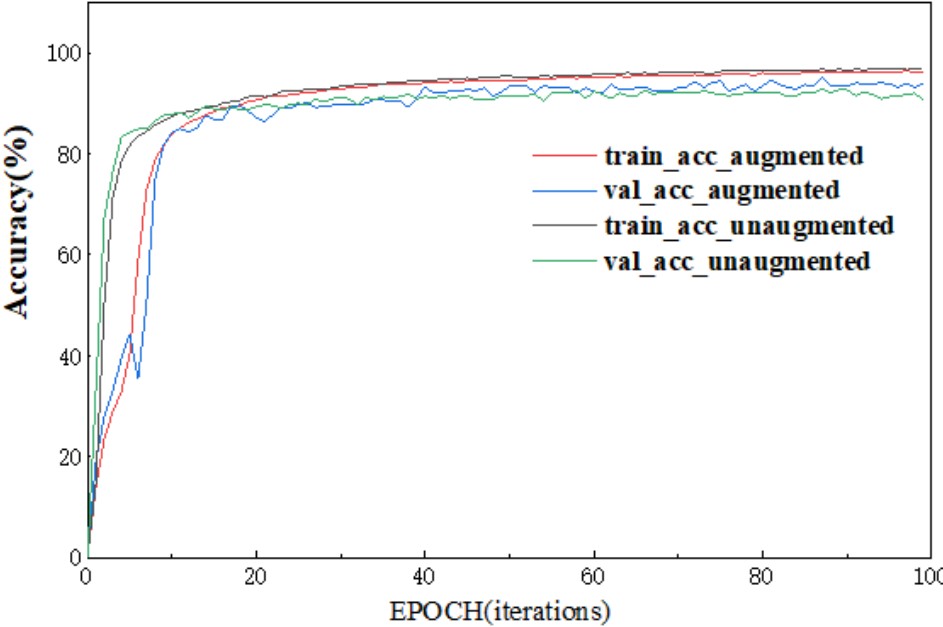

**Figure 13.** Train/validation accuracy on augmented/unaugmented dataset.

We then compare our model with the LSTM model and SDAE model used in Rimmer et al. [12] and the DF model used in Sirinam et al. [16]. The average training time and accuracy of the DF model proposed by Rimmer were compared under the unaugmented Wang dataset. From Figure 14, we can see that the Tor anonymous traffic recognition method proposed in this paper takes advantage of the parallelizing dilated convolutions model to effectively reduce the model parameters while maintaining a high anonymous traffic recognition accuracy. The convergence speed of the model is thus improved.

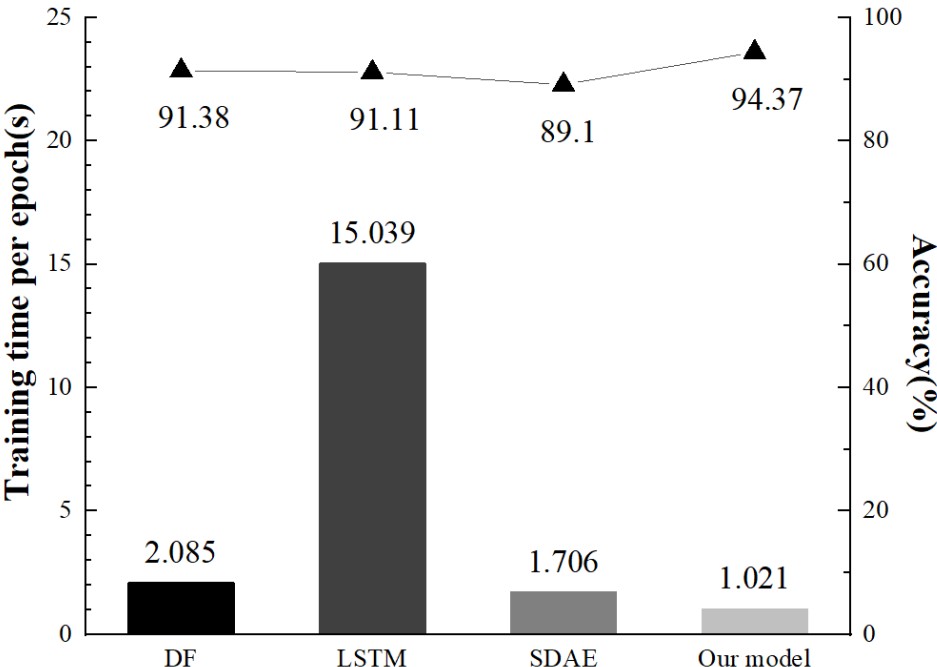

**Figure 14.** Training time and accuracy on different models.

### 4.4. Open-World Performance

In this section, we evaluate the Tor anonymous traffic identification model in a more realistic scenario (open world). In the open-world scenario, we need to distinguish between sensitive and non-sensitive pages for a given traffic flow. In the real world, the number of

non-sensitive pages is smaller than the number of sensitive pages and there is a category imbalance. A web fingerprinting attacker considers a situation where even a small FPR can lead to a large amount of anonymous traffic data being misclassified as sensitive pages when the sample set of monitored and non-monitored sites is large. In this situation, he often expects to reduce the FP value and hence the FTP rate, but this also leads to a reduction in the TP value and hence the precision. On the other hand, if an attacker wants to identify more potential users who visit sensitive pages, then he tends to expect to increase the TP value to regulate the potential users whereas increasing the TPR rate will also increase the FP value. We vary the threshold setting in the open-world experiment and fix the ratio of sensitive to non-sensitive pages in the validation set to 1:1 and the ratio of sensitive to non-sensitive pages in the test set to 1:2. We plot the ROC curves in our proposed model, the LSTM model proposed by Rimmer et al. [12] and deep fingerprinting model proposed by Sirinam et al. [16].

As shown in Figure 15, by constantly setting new thresholds, we can perform a TPR-FPR trade-off [13]. Our proposed model is near the top left corner of the figure which demonstrates that our model has a higher sensitivity and a lower false positive rate. In practical application scenarios, website fingerprint attackers can adjust the threshold settings in the open world according to their needs to achieve their personal desired recognition results.

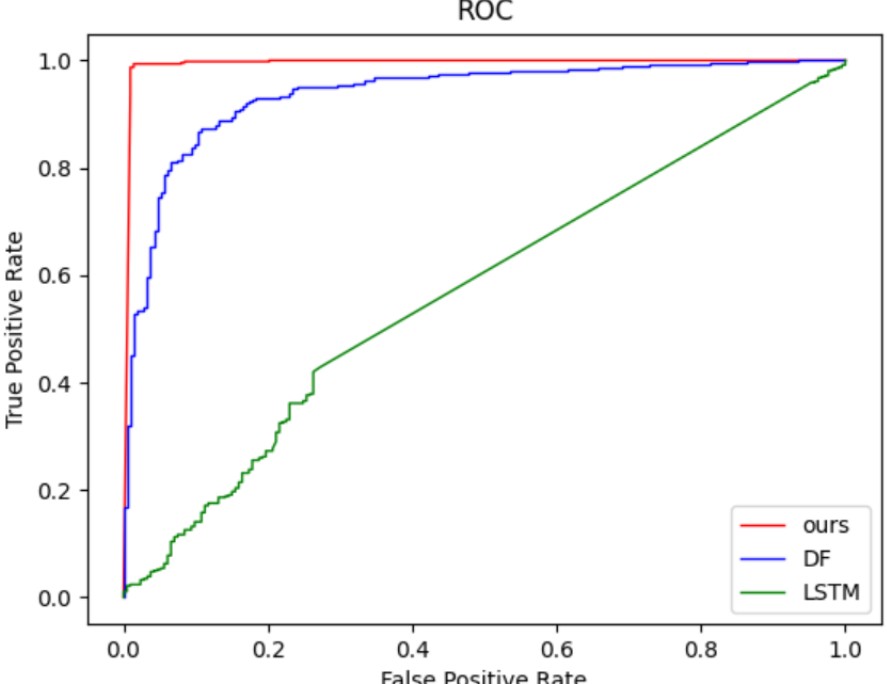

**Figure 15.** ROC on different models in the open-world setting.

Next, we adjust the ratio between the number of sensitive and non-sensitive pages in the test set, set the dichotomous threshold to 0.5 and plot the changes in each index for precision, accuracy and F1-score of our proposed model under different ratios, as shown in Figure 16.

As seen in Figure 16, the higher the ratio of monitored to non-monitored traffic in the test set, the higher the magnitude of all three metrics, with accuracy and F1-score improving faster and precision improving slower. Finally, we examine the performance of our proposed model, the LSTM model proposed by Rimmer et al. [12], the deep fingerprinting model proposed by Sirinam et al. [16] and the CUMUL model proposed by Panchenko et al. [22]. We kept the number of monitored and unmonitored sites as 1:2 for our experiments and the results are shown in Figure 17, where our proposed Tor anonymous traffic

identification model is slightly weaker than the DF model in terms of Accuracy metrics and achieves better results in terms of precision, F1-score.

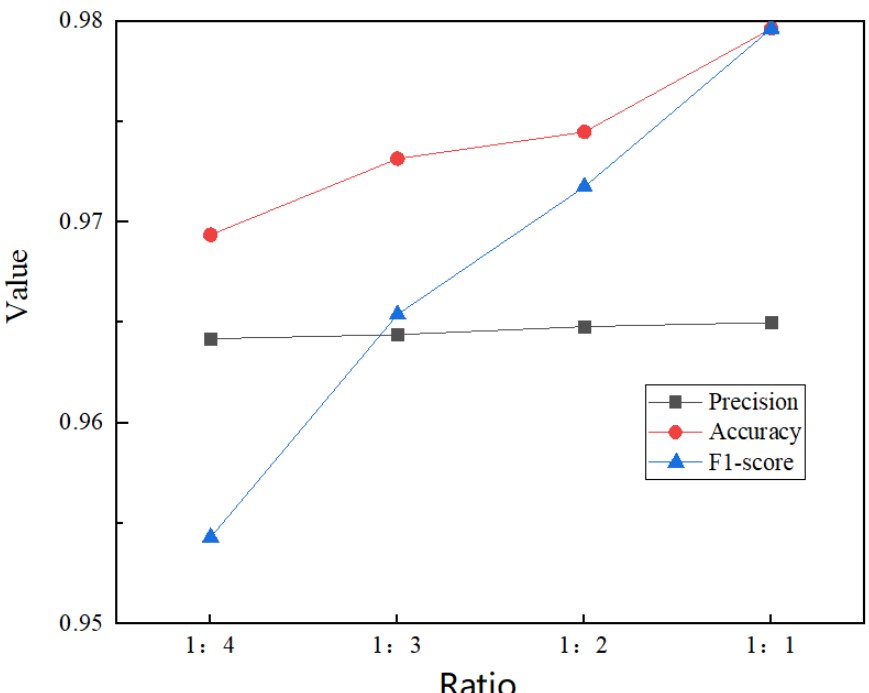

**Figure 16.** Changes in the ratio of sensitive sites to insensitive sites.

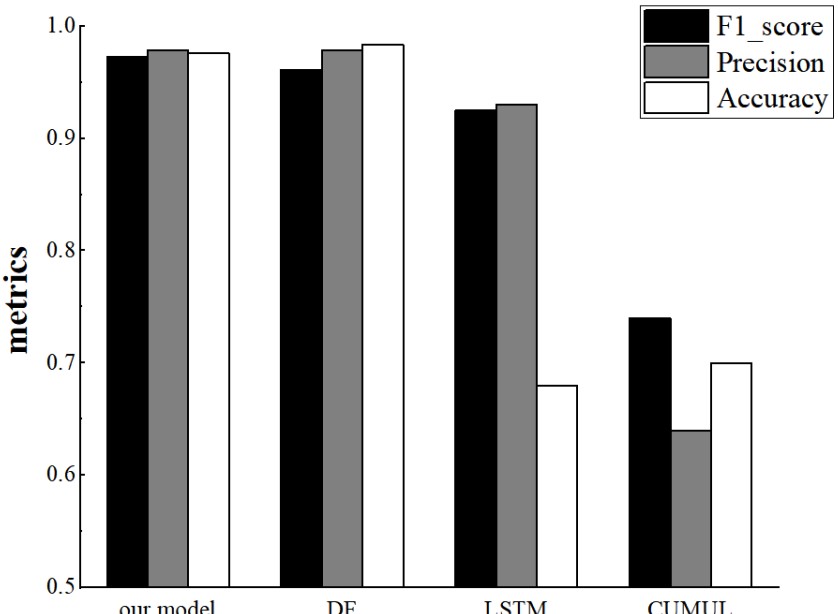

**Figure 17.** F1_score, precision and accuracy on different models in the open-world setting.

## 5. Conclusions and Future Work

In this paper, we propose a new Tor anonymous traffic recognition model that combines manual and automatic feature extraction methods, using parallelizing dilated convolutions to build a relatively lightweight network architecture. Directional and lengthwise data augmentation are performed for Tor anonymous traffic data continuity and bi-directionality. The proposed method is designed to alleviate the problems of slow network operation due to complex network architecture, inadequate feature extraction and biased fingerprinting owing to fast iterations of anonymous traffic. Our recognition accuracy achieves 94.37%

and our training time is cut by at least half (1.012 s per epoch compares to 2.085s per epoch in DF) using Wang dataset. As for data augmentation, we have a better fitting effect on the newly captured anonymous traffic data, which avoids the overfitting possibility of the model and enhances the robustness of our model.

The use of various deep learning methods for Tor anonymity identification has a wide range of promise but as identification techniques continue to evolve, increasingly more website fingerprinting defense strategies are being proposed [24–26], which poses a new challenge to our identification methods. Furthermore, in this paper we assume that Tor users only visit one web page at a time and do not make any content changes to simplify the scenario; whereas in practice, there are often more complex scenarios of multi-tabbed web visits. These more complex and variable scenarios will be further investigated in our subsequent work to better handle subsequent problems.

**Author Contributions:** Conceptualization, Y.L., M.C. and C.Z.; methodology, Y.L. and C.Z.; formal analysis, Y.L. and M.C.; investigation, Y.L., C.Z. and W.Z.; resources, Y.L. and C.Z.; data curation, Y.L. and M.C.; writing—original draft preparation, Y.L.; writing—review and editing, M.C. and C.Z.; visualization, Y.L. and C.Z.; supervision, M.C. and W.Z. All authors have read and agreed to the published version of the manuscript.

**Funding:** This research was funded by the Fundamental Research Funds for the Central Universities of People's Public Security University of China under Grant 2022JKF02009.

**Institutional Review Board Statement:** Not applicable.

**Informed Consent Statement:** Not applicable.

**Data Availability Statement:** The data that support the findings of this study are openly available from Tao Wang in https://www.cs.sfu.ca/~taowang/wf/index.html.

**Acknowledgments:** Useful suggestions given by Shufan Peng and Rongkang Xi are acknowledged.

**Conflicts of Interest:** The authors declare no conflict of interest.

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
