# Peer review of "Tor Anonymous Traffic Identification Based on Parallelizing Dilated Convolutional Network"

_applsci, doi:10.3390/app13053243_

Round 1
Reviewer 1 Report
The work is devoted to the development of a new model for recognizing anonymous traffic when using the Tor onion browser to protect websites using fingerprints. In this case, convolutional neural networks and their parallelization are used.
The article is of scientific and practical interest. However, for publication in the journal, the following remarks must be addressed.
1) At the beginning of the Materials and Methods section. The methodology should be improved by adding mindmaps/graphs and further clarifications.
2) No references to fig. 10
3) When reviewing the literature, it is necessary not only to list the work done by other authors, but also to analyze their result - why does it not suit us in the context of your further research?
4) At the end of section 2, formulate the purpose and objectives of the research based on the literature review, what do you want to improve?
5) There are no numerical results in the conclusions. What is the efficiency of your method, in % (and you need to provide a calculation or justification)).
6) Please provide screenshots that demonstrate your results. Please describe your experiment in more detail.
7) The schemes in Figures 2, 3, 4, 5, 6, 8, 9 look scattered, it is not clear how they are connected. On fig. 2 shows the general architecture. Please show (possibly highlight with color, label the number of the linked figure) - as parts of your method presented in fig. 3, 4, 5, 6, 7, 8, 9, what blocks they are shown in fig. 2? To make it clear how these drawings are related to each other.
8) Explain where formulas (4)-(12) are used? In what blocks of your schemes, algorithms? While this is not clear.
9) In addition, not everywhere there are references to the literature where these formulas are taken from, please add if these formulas are not your innovation.
10) Formulas (1) and (2) are vectors that follow from fig. 7, what next, where are they used? Where does formula (3) come from? Describe in detail. How does this affect the result?
To summarize, the article is hard to read, drawings, formulas and parts of the text are scattered. They need to be connected, to build the logic of the article, to show the movement from the input information to the result. I hope that working on my comments will improve the readability of your article.
Reviewer 2 Report
here are a few comments that need to be addressed:
- line 92: ... unencrypted data. (period)
- line 148: reference should be after "al." (also to correct in other locations in the paper)
- line 198: ... period of time
- line 370: switch on: (space after column)
- figure 9: 2st should be 2nd, 3st should be 3rd
- line 394: ... mapped by softmax to (should be something before reference)
- Section 3.4: content does not match the title. The crop on data and Figure 10 are not discussed in the text.
- line 406: sentence could be improved/clarified
- line 417: unexpected line break
- line 471: seems the crop approach is not discussed in the above sections
- line 492: unexpected character after LSTM
Reviewer 3 Report
1. Improve the summary with the following structure according to the recommendations of the IMRYD methodology
a. Identify the problem
b. Research objective
c. method used
d. Main result
e. Main Conclusion
2. Apply throughout the article the phases of the IMRYD methodology for a better organization
3. In figure 1 increase the attack sequence in two or three steps
4. Figure 2 modify to clearly visualize the content of the figure
5. Carry out the description of figure 3
6. On page 8 do not leave blank space
7. It is recommended to change all the references that are more than 5 years old so that the research has greater consistency.
Round 2
Reviewer 1 Report
1) Please enlarge the font in fig. 2, 3, 5, 6
2) I think that it is necessary to shorten the descriptive part in the review (section 2)
3) Still try to present the methodology in the form of mental maps. For example, as shown in the link: https://www.edrawsoft.com/3-basic-mind-map-types.html
You have everything described in detail, but there is a lot of text, but I would like to visually see the logic of solving the problem from its formulation to the solution.
4) In response to the comments, you provided a screen, but it is of poor quality, nothing is visible there. I recommend improving the quality of the screen and adding it to the article, as an attachment to the Appendix section.
5) In general, it would be good to shorten the text part, make the presentation more concise and structured.
